# Wrinkle force microscopy: a machine learning based approach to predict cell mechanics from images

Honghan Li[1,3], Daiki Matsunaga [1,3✉], Tsubasa S. Matsui[1], Hiroki Aosaki[1], Genki Kinoshita[1], Koki Inoue[1], Amin Doostmohammadi [1,2] & Shinji Deguchi [1✉]

Combining experiments with artificial intelligence algorithms, we propose a machine learning based approach called wrinkle force microscopy (WFM) to extract the cellular force distributions from the microscope images. The full process can be divided into three steps. First, we culture the cells on a special substrate allowing to measure both the cellular traction force on the substrate and the corresponding substrate wrinkles simultaneously. The cellular forces are obtained using the traction force microscopy (TFM), at the same time that cell-generated contractile forces wrinkle their underlying substrate. Second, the wrinkle positions are extracted from the microscope images. Third, we train the machine learning system with GAN (generative adversarial network) by using sets of corresponding two images, the traction field and the input images (raw microscope images or extracted wrinkle images), as the training data. The network understands the way to convert the input images of the substrate wrinkles to the traction distribution from the training. After sufficient training, the network is utilized to predict the cellular forces just from the input images. Our system provides a powerful tool to evaluate the cellular forces efficiently because the forces can be predicted just by observing the cells under the microscope, which is much simpler method compared to the TFM experiment. Additionally, the machine learning based approach presented here has the profound potential for being applied to diverse cellular assays for studying mechanobiology of cells.

[1] Division of Bioengineering, Graduate School of Engineering Science, Osaka University, 1-3 Machikaneyama, Toyonaka, Osaka 5608531, Japan. [2] Niels Bohr Institute, University of Copenhagen, Blegdamsvej 17, 2100 Copenhagen, Denmark. [3]These authors contributed equally: Honghan Li, Daiki Matsunaga. ✉email: daiki.matsunaga.es@osaka-u.ac.jp; deguchi.shinji.es@osaka-u.ac.jp

There is now growing evidence showing that cells sense mechanical cues in the surrounding microenvironment to regulate their functions such as proliferation, differentiation, apoptosis, and pro-inflammation[1–6]. In response to the mechanical cues, cells often adjust their cytoskeletal tension such that many of the mechanical information are translated into a level of inherent cellular traction forces, and in turn into intracellular signals regulating the related functions[3,7–9]. Traction forces, thus related to various cell functions, are generated by the activity of nonmuscle myosin II and actin filaments that determine cellular contractility[2,10–12]. Because these proteins work downstream of diverse signaling pathways, it is often difficult to predict how the force may change upon perturbations to particular molecules such as gene mutations and drug administration. Thus, technologies allowing for efficiently evaluating the cellular traction force are expected to enhance comprehensive understanding of the force-related pathways.

We previously developed a wrinkle assay, a modified version of the method originally reported by Harris and colleagues[13,14], in which the silicone substrates are spatially treated with uniform oxygen plasma to allow them to buckle upon the forces exerted by cells[15–17].

As the individual wrinkles are lengthened with the increase in the forces[18], the wrinkle length, detected for example by a machine learning approach[19], can be used as a measure of the relative change in the force caused by perturbations, such as specific gene mutations. This technology is promising in that these experiments are performed easily to potentially enable a high-throughput analysis on the force-related pathways or drug screening. For instance, the force measurements with dozens of different drugs can be done simultaneously by implementing the wrinkle assay in a multiwell plate[20]. However, the interpretation of the wrinkle length was not necessarily straightforward in terms of quantitatively measuring the magnitude and direction of traction forces. Although the geometrical information of wrinkles[21–23], such as wavelengths, would give an estimation for the force magnitude and direction, the geometry is still not enough to predict the local force distribution in a subcellular scale that is important to understand the cellular mechanotransduction and the morphological changes of cells. To overcome this limitation in quantification, here we describe a new machine learning system, wrinkle force microscopy (WFM), that converts the wrinkle information taken by a microscope into the actual cellular force distributions. For the initial training data, the cellular traction forces are obtained using the traction force microscopy (TFM), and we train the machine learning system with GAN (generative adversarial network) so that the network understands the way to convert the input microscope images to the force distributions from the training data. After sufficient training, the network can be utilized to predict the cellular forces just from the input images. The system would be a powerful tool to evaluate the cellular forces efficiently because the forces can be predicted by simply imaging the cells under the microscope, which is much simpler method compared to the TFM experiment.

## Results

**Full picture of WFM**. Our goal is to construct a machine learning system that can predict the cellular force distributions from the microscope image or the extracted substrate wrinkles. The full process can be divided into three steps as shown in Fig. 1. First, we culture the cells (A7r5; embryonic rat vascular smooth muscle cells) on a silicone membrane substrate and measure both the cellular traction force and the substrate wrinkles simultaneously. As shown in Fig. 1(a), the cellular traction forces are obtained using TFM[24,25], and cells generate wrinkles because the surface of PDMS (polydimethyl siloxane) layer is hardened by the plasma irradiation[16,19,20,26,27]. Second, the wrinkle positions are extracted from the microscope images as shown in Fig. 1(b) by using our SW-UNet (small world U-Net)[19], which is a convolutional neural network (CNN) that reflects the concept of the small world networks[28,29]. Third, the machine learning system utilizing GAN[30] is trained to understand a way to convert the microscope image, or the extracted wrinkle image, to the cellular force

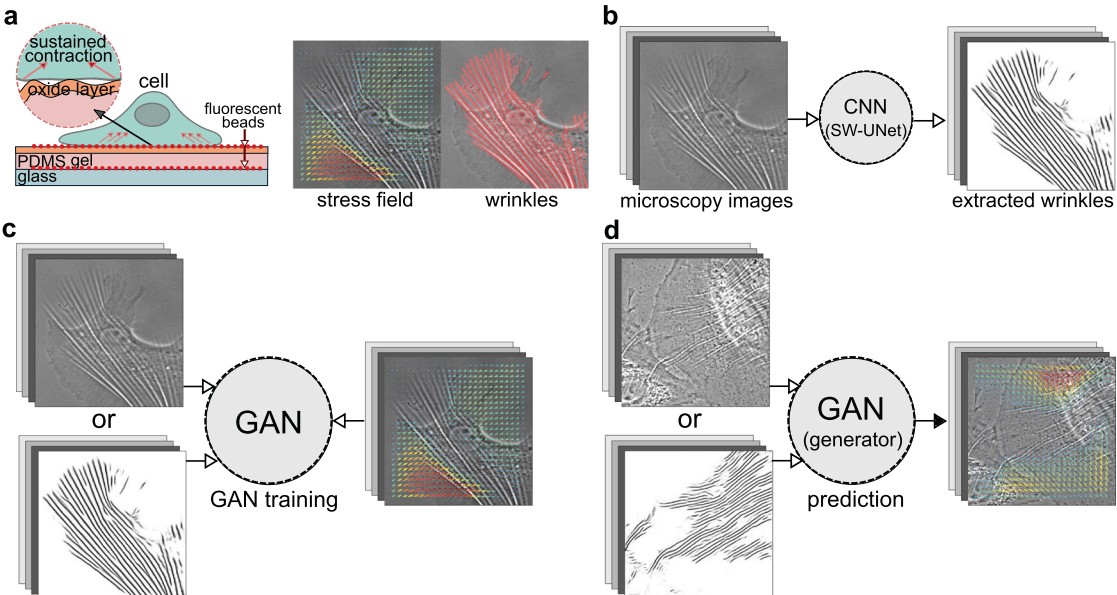

**Fig. 1 Overview of the methods and procedures that are utilized in the wrinkle force microscopy (WFM). a** Schematic of our experimental setup. A silicone membrane, which can evaluate the cellular force distribution (obtained by TFM) and the surface wrinkles simultaneously, is utilized in this work. **b** The surface wrinkles are extracted from the microscope images by using our machine learning system (SW-UNet). **c** The machine learning system (GAN) is trained to understand the relation between the input images (raw microscope image, or extracted wrinkle images) and corresponding output images (cellular force distribution). **d** After sufficient training, WFM can predict the force distributions only from the microscope images.

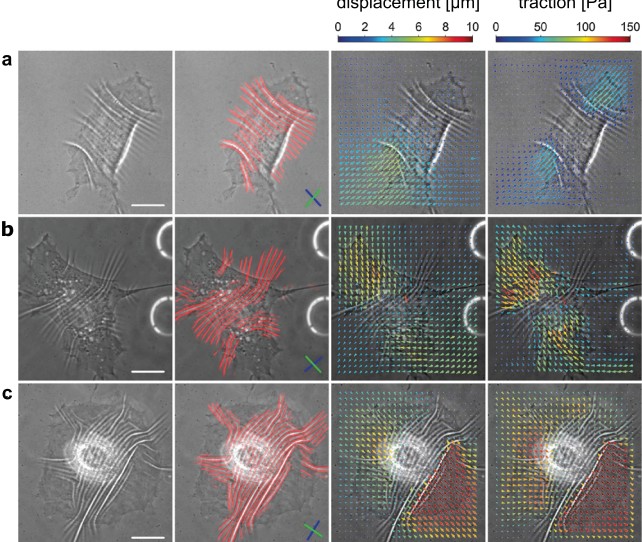

**Fig. 2 Three examples of the simultaneous measurement of wrinkles and traction forces.** Each column **a–c** describes (from left to right) raw image, wrinkles (red lines), displacement field and traction force field. The white scale bar in the first column images is 20 µm. The blue and green lines inside the second column images describe the principal direction of the wrinkle and the traction, respectively.

distributions as shown in Fig. 1(c). After the training, the network can be utilized to predict the cellular forces just from the microscope images as shown in Fig. 1(d).

**Simultaneous measurement of wrinkles and traction forces.** Before applying the machine learning system, we begin by considering the results of the simultaneous force and wrinkle characterization. Figure 2 summarizes representative results obtained by the experiment and analysis. Due to the pairwise inward pulling generated by cellular traction force (fourth column), the substrates exhibit displacements toward the cell center (third column). As the result of the contraction, the wrinkles emerge mostly underneath the cells (second column). When the cell size is small, the majority of wrinkles are aligned in a same direction as in Fig. 2(a), while they tend to point in different directions when the cell size is large and the traction is strong as in Fig. 2(c).

Figure 3(a) shows the probability distribution function (PDF) of the traction magnitude of $N \times M$ samples, where $N = 103$ is the number of the images and $M = 26 \times 26$ is the number of the force observation points. The average traction is $50.3 \pm 57.1$ [Pa] (mean ± standard deviation). Figure 3(b) shows that the wrinkle length has a positive correlation with the mean traction of the images, which is in agreement with our previous experimental measurements[26], where the relationship between the wrinkle length and applied force was experimentally investigated using microneedles. The mean traction is simply obtained by averaging the norm of the traction of the image as

$$\bar{f} = \frac{1}{M} \sum_m^M |\boldsymbol{f}_m| \qquad (1)$$

where $m$ is the index of the observation points. The wrinkle length is measured by counting the number of pixels after skeletonizing the wrinkle images[26]. The wrinkle extincts when the mean traction in an image is less than 10 Pa, which is comparable to the noise level or the resolution of the current TFM.

In order to analyze the principal direction of the traction, we construct a symmetric stress tensor for each image as

$$S_{ij} = \frac{1}{2M} \sum_m^M \left\{ n_j(\boldsymbol{x}_m, \boldsymbol{x}_0) f_i(\boldsymbol{x}_m) + n_i(\boldsymbol{x}_m, \boldsymbol{x}_0) f_j(\boldsymbol{x}_m) \right\} \qquad (2)$$

where $\boldsymbol{f} = (f_x, f_y)$ is the traction force, $\boldsymbol{r} = \boldsymbol{x}_m - \boldsymbol{x}_0$ is the relative vector from the image center $\boldsymbol{x}_0$ and $\boldsymbol{n} = \boldsymbol{r}/|\boldsymbol{r}|$ is the normal vector. By diagonalizing the tensor, we obtain the principal direction of the traction $\phi_s$ (shown in Fig. 2, second column with green lines on the bottom right corner) together with the corresponding principal traction magnitude $f_p$, from the eigenvalue that has the largest norm. At the same time, we obtain the principal direction of the wrinkles $\phi_w$ (also shown in Fig. 2, second column with blue lines on the bottom right corner) from the 2D-FFT (fast Fourier transform) image of the wrinkles: $\phi_w$ is an angle that is perpendicular to the direction that has a largest power spectrum. Figure 3(c) shows that the traction force is contractile ($f_p < 0$) and is almost linearly related to the wrinkle length (correlation $R = -0.82$ in a range $f_p < -5$ Pa). While earlier work[18,26] predicted a linear relationship between the traction force and the wrinkle length, analytical treatment[21] showed that the relationship should be quadratic. The linear relation that is measured here validates a more recent theoretical prediction based on far-from threshold theory of wrinkling[31] and is consistent with the experiments on droplets on polystyrene films[32]. As such, the wrinkles can be used for one qualitative marker or indicator for rough estimation of the cellular traction magnitude. Figure 3(d) shows that the two angles $\phi_s$ and $\phi_w$ are perpendicular most of the time. Since the wrinkle direction is perpendicular to that of the force dipoles, the wrinkle would be also practical to qualitatively predict the force directions, as previously done elsewhere[16,18]. Therefore, the length and direction of wrinkles provide a *qualitative* measure of the magnitude and direction of forces exerted by cells on the substrate, respectively.

Topological features of the wrinkle also give us insights into the force distributions. As shown in Fig. 4(a), some cells exhibit wrinkles in a single region (top) while the others show several separated regions of wrinkles (bottom). By categorizing the cell images (total 34 images) into two types as clustered patterns (9 images) and dispersed patterns (25 images), we summarize the difference in the force distributions in Fig. 4(b, c): the average force $\bar{f}$ and force isotropy $I$ are larger (significant difference only for $\bar{f}$: $p = 1.15 \times 10^{-2} < 0.05$, $I$: $p = 1.42 \times 10^{-1}$) for clustered pattern, where isotropy is evaluated as $I = |f_p/f_p^{\min}|$ and $f_p^{\min}$ is the smaller eigenvalue of $S_{ij}$. The wrinkles are generated by the pairwise forces that are transmitted via focal adhesions. When the wrinkles are dispersed, cells tend to be elongated, as seen in the pictures, and we conjecture that the cells might not be strong enough to generate continuous wrinkles as the clustered ones. This result suggests that the topological features of the wrinkles contain rich information about the force distributions.

Finally, the correlations between cellular properties and characters of contractile forces are summarized in Table 1. Since each variable has moderate positive correlations (~0.5) with all other variables, it can be concluded that the cells are circular when the size is larger, and both force magnitude and isotropy increase with the cell size. Taken together, these results show that wrinkles can provide qualitative information about the magnitude, direction, and distribution of traction forces that cells generate. Nevertheless, a quantitative relation between the wrinkles and traction forces is missing.

**Traction force prediction using GAN.** Next, we employ the machine learning-based approach to provide a *quantitative*

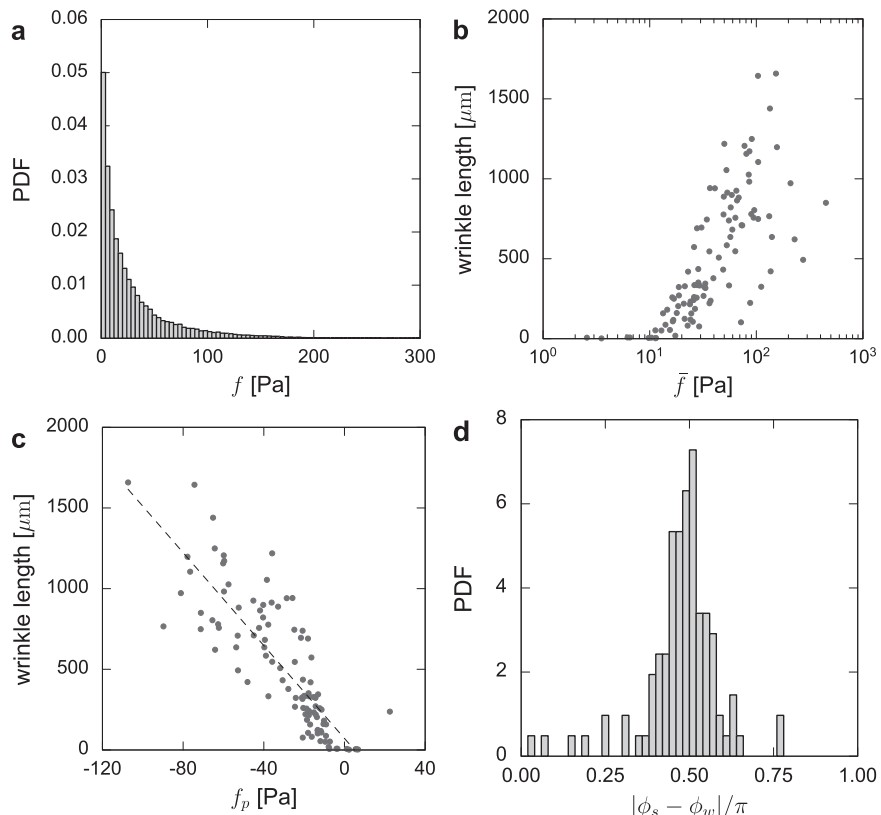

**Fig. 3 Quantitative analysis of the traction forces and wrinkles. a** Probability distribution function (PDF) of the traction magnitude. **b** The wrinkle length as the function of the mean traction $\bar{f}$. Note that the wrinkle length is evaluated by counting the number of pixels after skeletonizing the wrinkle images. **c** The wrinkle length as the function of the principal traction $f_p$. **d** Probability distribution function of the angle differences between the wrinkle direction $\phi_w$ and the traction $\phi_s$. The figure suggests that the direction of the wrinkles is predominantly perpendicular to the principal direction of the force.

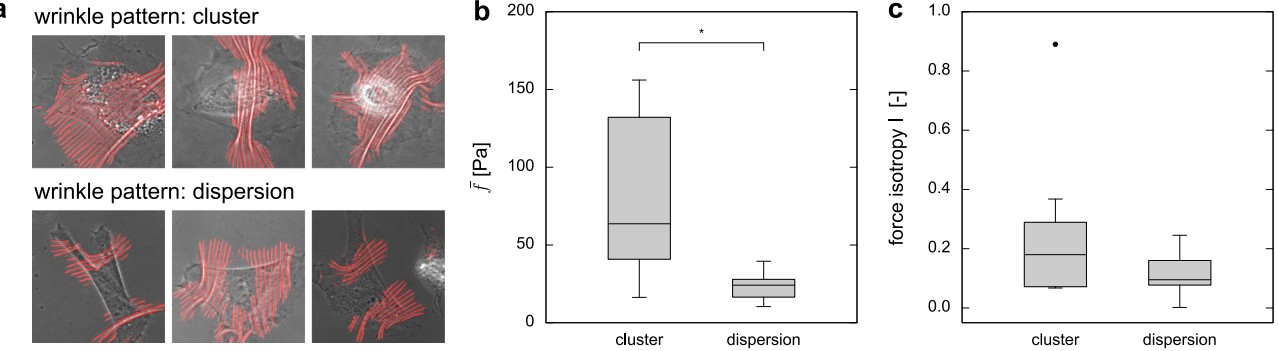

**Fig. 4 Topological features of wrinkles relate to force distributions. a** Three examples of cells that exhibit clustered wrinkles or dispersed wrinkles. **b** The average force $\bar{f}$ and **c** force isotropy $I$ for two groups of cells; cells that exhibits clustered or dispersed wrinkles. Note that asterisk symbols denote the significance $p < 0.05$ (*); two-sided unpaired $t$-test. The data numbers are $n = 9$ (cluster) and 25 (dispersion), respectively. The box plots describe minimum, first quartile, meadian, third quartile and maximum. Black dot is an outlier.

**Table 1 Correlations between cellular properties (area, roundness) and force characters (force isotropy, magnitude).**

|           | area  | roundness | force isotropy | force magnitude |
|-----------|-------|-----------|----------------|-----------------|
| area      | 1.000 | 0.626     | 0.584          | 0.610           |
| roundness | –     | 1.000     | 0.436          | 0.499           |
| isotropy  | –     | –         | 1.000          | 0.624           |
| magnitude | –     | –         | –              | 1.000           |

measure of the forces from the microscope images of wrinkles. We train the network and evaluate the performance of the force estimation using our GAN network. Figure 5 compares the predicted force distributions which were estimated by the three different methods. As also shown in Fig. S1, we trained the network with two different input images, the raw microscope images (second column in Fig. 5) and the extracted wrinkle images (third column), to compare the performance. We also evaluated the force distribution using a standard encoder-decoder type CNN and show the results in the fourth column. The figure shows that all the three methods reproduce approximately the same force

direction as the ground truth (first column), and the forces are perpendicular to the wrinkles.

Figure 6(a) compares the traction of ground truth $f_{x,y}^{\text{true}}$ and GAN prediction $f_{x,y}^{\text{predict}}$ (input image: microscope images), and it shows that the prediction is highly correlated with the experimental data. Note that we used $N = 252$ training image sets and 3 test images for the evaluation. The total error is calculated by averaging the error of 15 test images, which are obtained by

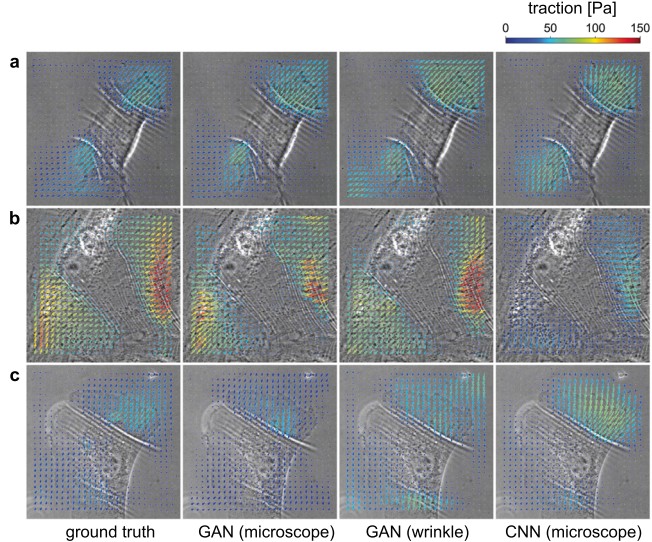

**a**

**b**

**c**

ground truth GAN (microscope) GAN (wrinkle) CNN (microscope)

**Fig. 5 Prediction of the traction forces from microscope images.** Each column **a**–**c** shows the result of the traction force predictions by using different methods (from left to right): ground truth, GAN prediction (input: microscope image), GAN prediction (input: extracted wrinkle images), and CNN prediction (input: microscope image). See further examples in Movies S1–S4.

repeating the evaluation 5 times with randomly selected different test images. The correlation coefficient $R$ averaging 15 test images is 0.84–0.88 for GAN and 0.82–0.84 for CNN as shown in Fig. 6(b), and it suggests that there are striking agreements. In order to further quantify the error in the force estimation, we introduce two errors: the error in the force magnitude $\varepsilon_f$ and the force direction $\varepsilon_\theta$. The error $\varepsilon_f$ is defined to evaluate the difference in the force magnitude between the ground truth $f^{\text{true}}$ and the prediction $f^{\text{predict}}$ as:

$$\varepsilon_f = \frac{1}{M} \sum_m^M \frac{|f_m^{\text{predict}} - f_m^{\text{true}}|}{f_m^{\text{true}}} \cdot \omega_m \tag{3}$$

where $M = 26 \times 26$ is the number of observation points, $\omega_m = f_m^{\text{true}}/\bar{f}$ is the weight function and $\bar{f}$ is the average force in the image which is defined in Eq. (1). Note that we introduce this weight function in order to put weight on the evaluation of large vectors rather than small vectors, which give huge errors even for small differences. Figure 6(c) shows that the error is 38–41% for GAN, and it has better performance compared to the encoder-decoder type simple CNN, which has an error of 50%. There is no obvious difference in the two input images (microscope images and wrinkle images), and this result indicates that the performance of the force estimation would not improve drastically by explicitly teaching the wrinkle position to the machine learning system. A comparison of the distributions between the prediction and ground truth is shown in Supplementary Fig. 1. Next, we evaluate the angle difference between the predicted force and the ground truth as

$$\varepsilon_\theta = \frac{1}{M} \sum_m^M |\theta_m^{\text{predict}} - \theta_m^{\text{true}}| \cdot \omega_m \tag{4}$$

where $\theta = \arctan(f_y/f_x)$ is the force direction. Figure 6(d) shows that the errors are 19–23° for GAN, and again shows better performance compared to conventional CNN ($\varepsilon_\theta = 24$–28°). Note that the spatial distribution of errors $\varepsilon_f$ and $\varepsilon_\theta$ is shown in Supplementary Figs. 2 and 3. As for a traditional CNN, the loss function is designed to measure the error between predicted

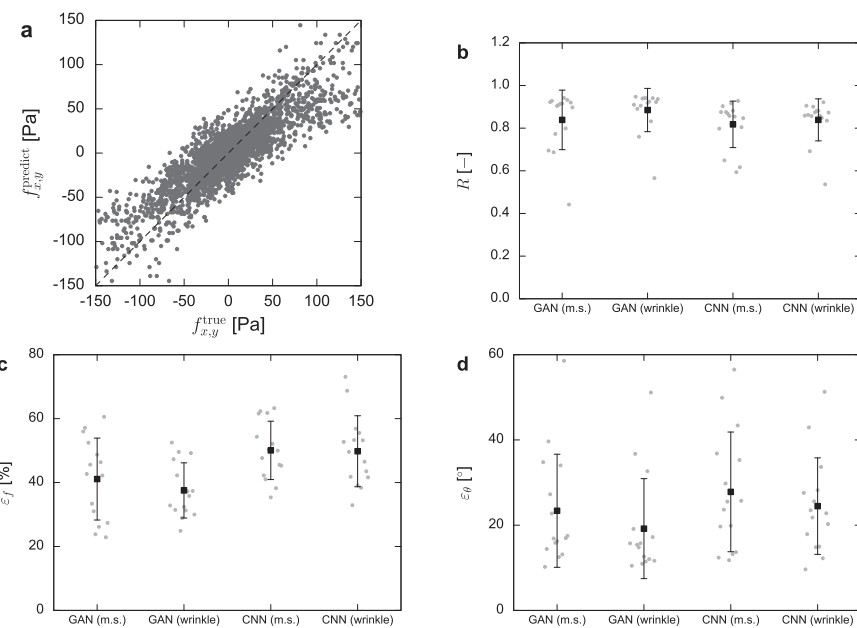

**Fig. 6 Quantification of the errors in prediction. a** Comparison of the ground truth $f_{x,y}^{\text{true}}$ and the predicted traction $f_{x,y}^{\text{predict}}$. Dashed line shows a condition $f^{\text{predict}} = f^{\text{true}}$. Note that we randomly reduced the number of data point 1/5 for the visibility. **b** Correlation coefficient $R$ between $f_{x,y}^{\text{true}}$ and $f_{x,y}^{\text{predict}}$. **c**, **d** Errors of the predicted traction compared to the ground truth data: **c** error in the traction magnitude $\varepsilon_f$ and **d** the traction direction $\varepsilon_\theta$. Note that bars show average errors ± standard deviations among the sample number $n = 15$, and gray dots indicate the original data. The description m.s. denotes the microscope images.

results and ground truth, and these criteria for the error are fixed during the training. In the case of GAN, the loss function can adapt to the specific problem dynamically because of the discriminator network, and this difference brings GAN a better score as shown in the figure. It is important to note that we have so far acquired a minimal required amount of training (original data: ~100) to demonstrate the novel concept of cellular force detection from microscope images. These errors will be minimized by increasing the number of the training data. As demonstrated above, WFM succeeded in estimating the force distribution just from the input images with limited levels of errors in real time. Supplementary movies S1–S4 further show the application of the proposed system in providing high throughout, real time measure of the traction force distributions during dynamic cell locomotion.

## Discussion

We proposed a machine learning-based system, WFM, that can predict cellular force distributions from microscope images. The full process can be divided into three steps. First, we culture the cells on a plasma-irradiated silicone substrate and measure both the cellular traction force and the substrate wrinkles simultaneously. The cellular traction forces are obtained using the TFM, while cells generate wrinkles on the underlying substrates. Second, the wrinkle positions are extracted from the microscope images by using SW-UNet. Third, we train the GAN system by using sets of corresponding two images, the force distributions and the input images (raw microscope images or extracted wrinkle images), as the training data. The network understands the way to convert the input images to the force distributions from the training. After sufficient training, the network can be utilized to predict the cellular forces just from the input images.

The relationship between morphology and biological functions of living creatures has long been an intense subject of research[33]. This topic has been investigated at the individual cellular level as well[27,34,35], in which cellular contractile forces were implicated in diverse functions including proliferation, differentiation, apoptosis, and tumorigenesis. Given its complicated nature, however, the general relationship associated with cellular forces remains poorly understood. In this regard, the technology described here has a potential to drastically advance research in this field as it allows for easier acquisition of the cellular traction force data. Indeed, compared to elongated cells, we found a stronger tendency for circular cells to produce contractile forces that are more isotropic and are higher in magnitude. Thus, WFM is expected to be applicable, in addition to drug screening as we discuss below, for extensively probing how the forces generated by cells is related to their functions including maintenance of morphological phenotypes.

It is instructive to discuss limitations and advantages of the proposed approach. In this paper we have presented data for a stiffness range 5.4−16.3 kPa. In principal, it is recommended to do the training again for different stiffness in most of the cases, except limited cases described below. As we have described in the Methods section "Wrinkle mechanics", conditions for the wrinkle generation would be identical if the stiffness ratio is the same for new experiment: $E_p^0/E_m^0 = E_p^1/E_m^1$, where $E_p$ and $E_m$ denote the Young's modulus for the elastomer and the oxidized surface layer, respectively. $E$ Therefore, the only thing one needs to modify is to multiply $E^1/E^0$ to the force that the WFM predicted for this case. When the stiffness ratio is not fixed $E_p^0/E_m^0 \neq E_p^1/E_m^1$, WFM cannot directly predict the force distribution since the wavelength $\lambda$ (Eq. [9]) and the critical strain for the wrinkle generation $\varepsilon_c$ (Eq. [11]) are different for this condition. Although it might be still possible to estimate the strain considering the condition

differences, it is still difficult to do the direct prediction as before. Additionally, since the wrinkle patterns are determined only from the traction distributions, we expect the same principle can be applied to other cell lines unless the cells have comparable sizes with those used in the training data. When the cell size is not comparable, the model should be trained once again since wrinkle patterns with similar length scale might not be included in the training data. Supplementary Fig. 4 also shows the distribution of traction force for MEF (Mouse embryonic fibroblast), which has a comparable size with A7r5 cells (training data).

Comparing with the TFM experiment (test data), the prediction using our system is highly correlated with the experimental data, with the averaged correlation coefficient of 0.84–0.88 and with 38–41% errors in the force magnitude prediction and angle errors 19–23° in the force direction. We expect that this error would decrease further by increasing the number of training images. The WFM would be a powerful tool to evaluate the cellular forces efficiently because the forces can be predicted just by observing the cells, which is much simpler method compared to performing the TFM experiment every time needed.

TFM is one of the most used methods to evaluate the cellular forces in mechanobiology study, but as the accuracy of the measurement depends on the successful acquisition of the reference positions of the micro-beads that are obtained by removing the cells after each of the experiments in conventional TFM, this method is limited in throughput. The novel GAN-based system proposed here overcomes this limitation as it provides the nearly same information, with the high levels of the correlations with the experimental data and the limited levels of the errors on the cell mechanics. Importantly, this is achieved only from the still images that are acquired by plating the cells on the silicone substrate without taking care of the reference as the substrate surface is known to become planar again upon the absence of the cellular forces in a reversible manner. Given that early stages of drug screening require testing of a massive number of candidate compounds[20], our system, with the potentially high-throughput data analysis capability, will be useful particularly in such screening studies. It is important to note that our WFM is not the one that essentially competes with TFM, but the huge advantage of the proposed system is focused on its capability to provide data equivalent to the TFM (with a level of the errors) and thereby circumvent performing the TFM that needs considerable technical care. Rather, because the machine learning system depends on the training data, further innovations in TFM such as super-resolution imaging[36,37] are potentially introduced to our system to synergetically output more sophisticated data. Thus, our approach presents a versatile framework that integrates the sophisticated experimental techniques and the efficient measurements.

## Materials and Methods

**Step 1: simultaneous measurement of traction forces and wrinkles**. Based on our previous studies[16,19,20,26,27], we prepared the substrate that can reversibly generate wrinkles upon application of cellular forces. First, a circular cover glass is treated with oxygen plasma (SEDE-GE, Meiwafosis) to hydrophilize the surface and is desiccated after fluorescent micro-beads (0.2 μm in diameter, carboxylate yellow-green fluorescent beads; Invitrogen) in water solution are distributed on the surface. Second, parts A and B of CY 52-276 (Dow Corning Toray) are mixed at a weight ratio of 1.2:1 and poured onto the cover glass to create a PDMS layer with a height of 30–40 μm. Third, the cover glass is placed in a 60 °C oven for 20 h to cure the PDMS. Fourth, oxygen plasma is applied uniformly along the surface of the PDMS layer to create an oxide layer that works as the substrate for cell culture. Finally, the substrate is coated with 10 μg/mL collagen type I solution for 3 h.

For the TFM measurement, fluorescent micro-beads are attached to the substrate surface as position markers to measure the substrate deformations. The beads need to be firmly adhered to the surface so that cells would not move the beads due to endocytosis. In this work, the covalent bonding between the surface and the beads of 0.001% v/v are performed by following two steps: (i) silane coupling of the substrate surface using 3-Aminopropyltrimethoxysilane and (ii) the

covalent bonding formation due to carbodiimide. The beads adhered on the glass surface are monitored to keep the reference position even after removing the cell using 0.25% Trypsin (Trypsin + 1mm mmol/I EDTA-4Na solution; Fuji Wako Pure Chemical Corporation).

**Cell culture and microscope setup.** A7r5 cells (purchased from ATCC) were maintained at 37 °C in a stage incubator (INUF-IX3W; Tokai Hit) under a humidified 5% $CO_2$ incubator. An inverted microscope (1X73; Olympus) with a confocal unit (CSU10; Yokokawa Electric) and oil immersion lens (phase contrast, UPlanFLN 60x/1.25 Oil Iris Ph3, Olympus Corporation) are used to capture the cells and fluorescent beads. During the experiment, DMEM(L)+10% FBS+Penicillin-Streptomycin (Fuji Wako Pure Chemical Corporation) is used as the culture medium.

Note that the proliferation (Supplementary Fig. 5) and migration (Supplementary Fig. 6) with three different substrates (CY with mixed ratio 1.2:1, CY with 1.0:1 and glass) are evaluated, and we confirmed that the wrinkles have no fatal or harmful effect on the cell nature.

**Traction force microscopy (TFM).** The software ImageJ/Fiji[38] and its plugin FTTC (Fourier transform traction cytometry)[39,40] are used to evaluate the force field from the displacement field. The substrate is considered as a soft elastic isotropic material that follows the linear elastic theory. First, the displacement of the substrate surface $u$ is measured by tracking the movement of the fluorescent beads using PIV (particle image velocimetry). The spatial resolution of the force distribution is 3.44 μm × 3.44 μm. Second, the traction field is obtained from the displacement field by solving the governing equation for the elastic halfspace[41,42] given by

$$u(x) = \int_S G(x,y)t(y)dS(y) \tag{5}$$

where $t$ is the traction force, $x$ and $y$ are the positions of the displacement and the traction force, respectively. $G$ is the Green's function that is given by

$$G(x) = \frac{1+\nu}{\pi E r^3} \begin{pmatrix} (1-\nu)r^2 + \nu r_x^2 & \nu r_x r_y \\ \nu r_x r_y & (1-\nu)r^2 + \nu r_y^2 \end{pmatrix} \tag{6}$$

where $E$ is the Young's modulus, $\nu$ is the Poisson's ratio, $r = (r_x, r_y) = x - y$ is the relative position vector and $r = |r|$. The software FFTC solves Eq. (5) in the Fourier space, which is given by

$$\tilde{t} = (G^T G + \lambda^2 I)^{-1} G^T \tilde{u} \tag{7}$$

where tilde symbols denote the variables in Fourier space, $\lambda$ is the regularization parameter[42] and $I$ is the unit tensor. In order to evaluate the optimal parameter $\lambda$ for the Tikhonov regularization, the L-curve criterion[42,43] is applied. Note that $E$ is experimentally determined[16] to be 5400 Pa and $\nu$ is assumed 0.5 (incompressible) that is a typical value for PDMS material.

**Step 2: Wrinkle extraction.** We use a method SW-UNet[19], which is a CNN based on U-Net[44] to extract wrinkle patterns from the microscope image as shown in Fig. 1(b). As the training data, we prepare 236 sets of corresponding two images (microscope image and manually labeled wrinkle image). The number of data is increased to 2596 by using the image augmentation techniques. We used NVIDIA Titan RTX to accelerate the training process, and the Adam optimizer is utilized. Note that the procedure until Step 2 was already developed and utilized in our previous papers[19,20], and GAN-based force estimation is newly presented in this work.

**Step 3: prediction of traction force based on GAN-based system.** Assume that we have $N_o$ sets of corresponding images and data; the input images (microscope images, or extracted wrinkle images) and the force distributions as shown in Supplementary Fig. 7. We effectively have the number of training data set $2N_o$ because the wrinkle image has only 1D information at each pixel (intensity $I(x,y)$; see also images in Fig. S1) while the force distributions have 2D information (2D force, $f(x,y) = \{f_x(x,y), f_y(x,y)\}$). We designed the network to evaluate the cellular force only for a single axis at one time and focus only on the $x$-directional force at each evaluation. An input image $I_i$ is used as the training data set $(I_i, f_x)$ and $(I_i', f_y)$ where $I_i'$ is an image that rotates $I_i$ by 90 degrees.

The force distributions are converted to gray scale images that have intensities

$$I(f_d) = a \arctan\left(\frac{f_d}{b}\right) + I_{\mathrm{mid}} \tag{8}$$

where $a = 81.2$, $b = 50.0$, $I_{\mathrm{mid}} = 255/2$ are the coefficients for the conversions, and $f_d$ is the components of the force $d = x, y$. The force distributions in grayscale, which are generated from the test images, can be converted back to the force using this equation. As the training data, we prepared $N = 252$ sets (63 original images) of corresponding two images. Note that 63 images are acquired from experiments on five different days and seven different dishes. We increased the number of training data by rotating the images: $63 \times 4 = 252$. Python programs using

TensorFlow and training data are provided in the following Github link https://github.com/Minatsukiyoshino/Wrinkle_force_microscopy.

**Wrinkle mechanics.** The wrinkle mechanics[22] can be discussed by simplifying the current substrate as an elastomer (Young's modulus: $E_m$) with a stiff oxidized surface layer (Young's modulus $E_p$, thickness $h$). Note that the Poisson's ratio $\nu$ of both materials are considered to be the same for simplicity. Substrate compression results in a sinusoidal buckling with a wavelength $\lambda$ and amplitude $A$ as

$$\frac{\lambda}{h} = 2\pi \left(\frac{E_p}{3E_m}\right)^{1/3}, \tag{9}$$

$$\frac{A}{h} = \left(\frac{\varepsilon}{\varepsilon_c} - 1\right)^{1/2} \tag{10}$$

where $\varepsilon_c$ is the critical strain that the wrinkle emerges, which is defined as

$$\varepsilon_c = -\frac{1}{4}\left(\frac{3E_m}{E_p}\right)^{2/3}. \tag{11}$$

The wavelength in our present system typically in a range $\lambda = 2-4$μm while the thickness of the oxidized layer $h$ is assumed to be in an order of 100 nm[45]. By assuming the ratio as $\lambda/h \sim 30$, we can give estimations on the stiffness ratio as $E_p/E_m \sim 300$ and the critical strain as $\varepsilon_c \sim -0.01$. Note if one needs to observe 20 wrinkles for each cell with a size ~100 μm, the stiffness ratio should be $E_p/E_m \leq 1500$ from Eq. [(9)]. Since the maximum strain is nearly $\varepsilon = -0.05$ in the experiment, the maximum amplitude of wrinkles is assumed to be $A \sim 200$ nm.

If the substrate stiffness is different from the trained condition, the system needs to be trained once again with a new set of data. However, the trained system is still applicable when the stiffness ratio $E_p/E_m$ of a new substrate is the same as the trained condition since the criteria for wrinkle generation would be identical, as shown in Eqs. [(9)] and [(11)]. By multiplying the ratio of Young's modulus, we can estimate the force distribution for this case.

**GAN structure.** Our goal is to convert a physical quantity (wrinkle geometry) to another physical quantity (force distribution). Even though the mechanical formulation between the two quantities is given, it is not necessarily straightforward to solve this inverse problem because of its complex nonlinear dynamics[21,46]. Instead, we achieve this purpose by training our machine learning system to understand the underlying mechanical rules. Considering this conversion of the physical quantity as an image "translation", we utilize GAN (generative adversarial network)[47] in this work.

GAN mainly consists of two networks, generator $G$ and discriminator $D$ as shown in Fig. S1, and the network is trained by a competition of two networks. Goal of the generator $G$ is to generate fake images (fake force distributions $G(x)$) from the input images $x$ (microscope images or extracted wrinkle images) and tries to mimic the real images $y$ (real force distributions), while the discriminator $D$ tries to distinguish true and fake images from the group of images. As the training proceeds, the generator learns how to produce fake images that are difficult to be distinguished by the discriminator from the real images, and the discriminator learns the rules to distinguish true/fake images. Once the training is completed, the trained generator $G$ can now be used as the translator to predict the force distribution from the input images $x$ even for test images, which were not included in the training process.

In the present work, we design the generator $G$ with a form of encoder-decoder which is based on U-Net[44], and Markovian discriminator (PatchGAN)[48] is utilized as the discriminator $D$. The generator $G$ converts the input images $x$ to the fake force distribution images $G(x)$. The images of force distribution, $y$ or $G(x)$, and the input images $x$ are concatenated into a single image as the input of the discriminator. The two networks $G$ and $D$ are trained based on the labels of real/fake, and we utilize the loss function $\mathcal{L}$ that is used in pix2pix[30]:

$$\mathcal{L}(G,D) = \mathbb{E}_{x,y}[\log D(x,y)] + \mathbb{E}_x[\log(1 - D(x, G(x)))] + \lambda L_1(y, G(x)) \tag{12}$$

where $\mathbb{E}$ is the expected value, $L_1(G)$ is the L1 distance between generated images $G(x)$ and ground truths $y$ and $\lambda = 100$ is the weight for the L1 term. The first term $\mathbb{E}[\log D]$ denotes the expected probability that the discriminator categorizes $y$ as the real data, while the second term $\mathbb{E}[\log(1 - D(x, G(x)))]$ denotes the probability that the discriminator categorizes the generated image $G(x)$ as the fake data. The goal of the generator $G$ is to minimize $\mathcal{L}$ while the discriminator $D$ tries to maximize it.

We use the training parameters as follows: 100 training epochs (batch size = 1), $\varepsilon = 0.0002$ learning rate for generator and discriminator, the parameters $\beta_1 = 0.5$ and $\beta_2 = 0.9$ are used for the Adam optimizer. The whole learning process is again accelerated by Nvidia Titan RTX.

**Statistics and reproducibility.** 66 cell images and corresponding force distributions are collected from the experiments of 5 different days and 7 different independently prepared dishes. The test data (3 cells) are randomly selected from the dataset, and all others (63 cells) are utilized as the training data. By changing the test data, the training and error evaluation were repeated 5 times. The sample numbers $n$ for each analysis are listed in the figure caption. The $p$ value of the unpaired $t$-test is evaluated with Microsoft Excel.

**Reporting summary**. Further information on research design is available in the Nature Research Reporting Summary linked to this article.

## Data availability

Source data for the graphs and charts in the figures are available in Supplementary Data 1, and any remaining information can be obtained from the corresponding author upon reasonable request. The machine learning code along with the training dataset that generated all the cell mechanical information in Fig. 5 and Fig. 6 can be found on GitHub (https://github.com/Minatsukiyoshino/Wrinkle_force_microscopy/).

## Code availability

The source codes of Wrinkle Force Microscopy (WFM) are publicly available on GitHub (https://github.com/Minatsukiyoshino/Wrinkle_force_microscopy).

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

## Acknowledgements

This work was supported by JSPS KAKENHI Grant Number 18H03518, 20K14649 and 21J22170, ACT-X JST (Grant No. JPMJAX190S), and Multidisciplinary Research Laboratory System for Future Developments (MIRAI LAB). AD acknowledges support from the Novo Nordisk Foundation (grant no. NNF18SA0035142), Villum Fonden (grant no. 29476), Danish Council for Independent Research, Natural Sciences (DFF-117155-1001), and funding from the European Union's Horizon 2020 research and innovation program under the Marie Sklodowska-Curie grant agreement no. 847523 (INTERACTIONS).

## Author contributions

H.L. and D.M. contributed equally as co-first authors. H.L. and D.M. developed the machine learning system, and K.I. supported the implementation. H.A., G.K., T.S.M., and

S.D. designed and worked on the cell experiments. H.L., H.A., and D.M. analyzed the experimental data. D.M., A.D., and S.D. conceived the idea of simultaneous TFM with wrinkle extraction. D.M. and A.D. analyzed the physics. H.L., D.M., A.D., and S.D. wrote the article and designed the research.

## Competing interests

The authors declare no competing interests.
