## [Peer Review File · Communications Biology]

Reviewers' comments:

Reviewer #1 (Remarks to the Author):

In this work, Honghan Li and colleagues describe an analysis pipeline showing how Deep Learning approaches can be used to predict the forces generated by cells from wrinkle force microscopy images. Such an approach promises that it would 1) enable fast traction force measurements, 2) avoid the phototoxicity associated with classical TFM approaches, and 3) easy to implement. Overall the work is of interest for the biological community interested in analysing how cells apply force on their substrate. My main concern is that the results are not described sufficiently to be reproduced by others. Also, this article is very much a proof of principle article, and no new biological phenomenon is described here.

The authors do not provide the code, or the training dataset(s) used to produce the deep learning models presented here. The authors also do not provide the model themselves. I believe that these should be provided alongside the paper so that readers can have access to these critical materials. To use the strategy described here, one will need to reimplement the algorithm used, generate their training data and train them. This will drastically limit the usability of the method described here. Also, no data availability statement appear to be available. How were the Deep learning model implemented? No information concerning the programming language or essential library used are available.

Regarding the approach itself, does the wrinkles affect cellular behaviour? or the amount of forces generated by the cells? Substrate patterning is well known to affect cellular functions (migration, proliferation and cell fate). How prominent are the wrinkles?

What is the force threshold for wrinkle formation? What is the sensibility of this approach? It is also unclear how this approach can resolve smaller forces that may arise at different angles than the prominent wrinkles.

This approach uses PDMS for force measurement. What are the stiffness range that can be used to visualise wrinkles? Should a new Deep Learning model trained for each stiffness to be analysed? On the same point, Should a Deep learning model be trained for each cell type to be imaged? Can the authors demonstrate that this is not the case?

What is the resolution of the approach used here? The TFM maps used are of very low resolution. Can this approach also be used to resolve smaller details?

The authors show that they obtain excellent predictions using their approach. It is, however, unclear how these analyses were done as no details are provided. How many images were analysed? Were these images part of the training dataset used to train the Deep Learning network?

While I do not think that the authors should discover new biological phenomena, they should demonstrate that this approach can be used to observe meaningful changes in force transmission. For instance, show that forces are reduced following a drug treatment.

The discussion is very much a summary of the study itself. I would recommend that the authors also discuss the limitations of the approach described here and its possible uses.

Minor:

Fiji software is not referenced.

It would be useful to provide the actual images used to train and generated by the GAN. Not just images with force vectors.

Reviewer #2 (Remarks to the Author):

The authors in this manuscript combine the well-recognized method of TFM with the much older described wrinkling imaging on silicon surfaces to propose an easier method to measure traction

exerted by the cells on a deformable substratum using machine learning.

The goal is to propose a much simpler method based on only phase contrast microscopy imaging. For this, authors seeded cells on silicon coated with fluorescent beads to allow performing on the same field traction force and wrinkling imaging. Then based on TFM data they trained neuronal network to extract traction fields. Although I am not in a position, as biologist, to evaluate what is new here compared to ref 18 and 19, I have some concerns about:

- the spatial resolution
- the sensitivity of the methods
- it possible or not application to cell layers
- the fact that silicon wrinkling modifying the layer in 3D may affect TFM extraction

These major points are not addressed here and they need at least to be evaluated.

Also, as authors say, this would be mostly interesting for screening, high throuput, but no proof of principle for its application in this field is provided.

Reviewer #3 (Remarks to the Author):

Li et al. provide an elegant method to extract quantitative absolute traction force measurements from wrinkle images using deep learning approaches. The absolute measure of forces is typically done using Traction Force Microscopy (TFM), which requires the imaging of fluorescent bead displacement which can then be converted into absolute forces from the knowledge of the mechanical properties of the substrate. An associated method uses the observation of substrate buckling (leading to the observation of wrinkles) which only requires the acquisition of brightfield or phase contrast of the cells' substrate but is less quantitative with respect to absolute force measurements.

Here, the authors generate a paired dataset of phase contrast images of wrinkles and TFM data, the latter being able to generate force maps. Then they used this data to train a conditional GAN neural network to predict force maps from such phase contrast images. This has the advantage to only require the acquisition of phase contrast images without the need for a reference image (as it typically required by TFM). The authors show nice quantifications of the performance of the approach on test dataset that show a broad agreement with ground truth and demonstrate the validity of the method.

Deep learning constitutes a great approach to perform complex transformation of data such as wrinkle images into force maps. The authors rightly explain that the information is there but may be difficult to extract in a quantitative and spatially resolved manner as TFM does. What I am not sure about is how it compares to previous efforts to convert wrinkle images into force measures. The authors mention that wrinkle length and direction constitute two measures that relate to amplitude and direction of the forces but has there been any past efforts to convert these into force maps using deep learning or otherwise?

The authors should discuss this in a bit more details, perhaps in introduction, and if no work has ever intended to do this quantitatively successfully, that will only strengthen the case for using deep learning and the present method.

The authors quantify errors compared to ground truth TFM data from the ensemble distribution of force magnitudes and angles. They show an agreement of the best performing method (GAN from wrinkles) within about 30% and mention that more training data would improve that. The angular errors are within 20 degrees. I would like to see a little more characterisation of these errors as they form the basis of the method and would help understand the caveats compared to TFM (here considered as gold standard).

So here, I think it would be useful to see the actual distributions of force amplitudes and force angles from both predictions and ground truth to show precision and potential biases. This could also be briefly discussed in my opinion.

Additionally, since the approach is meant to spatially resolve the forces, it would also be useful to show spatial error maps (difference or root square error, RSE, or similar) of both magnitude and angle for the test datasets, as is commonly done for validation of deep learning producing images. This would also potentially highlight issues of where the errors come from mostly and how they

relate to certain spatial features. This would be a nice complement to the correlation curve shown in Fig. 5a.

I also have a concern about generalisation of the work. Although the authors show nicely that there is a decent agreement with TFM from the test data. It looks to me that all the dataset (both training and test) were acquired on the same day from the same dish. I would like to see whether the approach would generalise to a couple of test datasets acquired on a different day from a different dish, that were not present in the training data. This would constitute the ideal test dataset here. Biological variability as well as variability of how the substrate may be made can cause variability that may throw off the model and produce poor quality predictions. This is not uncommon.

Reproducibility is also an issue here since neither the data nor the code for Deep Learning has been made available freely and directly. This is to me an important aspect that's missing and essential for transparency.

Minor comments:

Box plots in Fig. 5 are not defined.

Temporal information of the time-course data shown in the movies are not indicated anywhere.

Typo: "plasma-irradiated silocone"

Reply to reviewer's comments

We gratefully acknowledge the constructive comments and suggestions of the referees and the editor. The responses to each reviewer's comments are listed below.

Before replying to each comment, we would like to note that we reduced the number of the training dataset from 332 to 252. In the submitted version, we included same cells at different time (few hours difference; 20 cells) in order to increase the number of the training data. In the revised manuscript, we decided not to include those cells in order to show the robustness of our method. Note that it is common to do "data augmentation" in order to increase the number of training data in the field of machine learning, since the performance increase with the data amount. Instead of the image processing techniques that are usually utilized to increase the training images, we used same cells at different time to increase the data, in the submitted version. Since we noticed that the WFM works perfectly even without those same cells, we decided to remove those dataset from our training.

REVIEWER #1:

In this work, Honghan Li and colleagues describe an analysis pipeline showing how Deep Learning approaches can be used to predict the forces generated by cells from wrinkle force microscopy images. Such an approach promises that it would 1) enable fast traction force measurements, 2) avoid the phototoxicity associated with classical TFM approaches, and 3) easy to implement. Overall the work is of interest for the biological community interested in analysing how cells apply force on their substrate. My main concern is that the results are not described sufficiently to be reproduced by others. Also, this article is very much a proof of principle article, and no new biological phenomenon is described here.

Response to Comment 1:

We share the same concern about the importance of reproducibility and as detailed below have now appended all the data and the codes alongside the paper, and have made them freely accessible on github.

Comment 1-1:

The authors do not provide the code, or the training dataset(s) used to produce the deep learning models presented here. The authors also do not provide the model themselves. I believe that these should be provided alongside the paper so that readers can have access to these critical materials. To use the strategy described here, one will need to reimplement the algorithm used, generate their training data and train them. This will drastically limit the usability of the method described here. Also, not data availability statement appear to be available. How were the Deep learning model implemented? No information concerning the programming language or essential library used are available.

Response to Comment 1-1:

We have uploaded the code and the data to an open-access workplace, at https://github.com/Minatsukiyoshino/Wrinkle_force_microscopy. We use Python and TensorFlow for this program. These details are also now added to the Methods section of the revised manuscript.

Comment 1-2:

Regarding the approach itself, does the wrinkles affect cellular behaviour? or the amount of forces generated by the cells? Substrate patterning is well known to affect cellular functions (migration, proliferation and cell fate).

Response to Comment 1-2:

It is well-established that both the substrate stiffness (which affects wrinkles pattern) and the contact guidance mechanism (induced by wrinkles) affect the migration speed of the cells (Dokukina and Gracheva, Biophysical Journal, 2010). To test this in our experiments, we evaluated the proliferation (Figure S2) and migration (Figure S3) with three different substrates (CY with mixed ratio 1.2:1, CY with 1.0:1 and glass). Although the cell proliferation is slightly slower for glass substrate, there is no qualitative difference in the growth rate. On the other hand, the migration velocity is different for three cases, as expected (Dokukina and Gracheva, 2010).

In summary, although there are difference in the cell movements due to the substrate differences, there is no fatal or harmful effect on the cell nature. We have added these results and the corresponding discussion on the impact of the substrate stiffness to the revised manuscript.

In Supplemental materials:

added) Figure S2 and S3

Comment 1-3:

How prominent are the wrinkles?

Response to Comment 1-3:

This is a very good comment that can be addressed based on the following two points:

- We added a new section “Wrinkle mechanics” in the materials and method. Based on the existing theories of non-linear wrinkle formation on elastic substrates, the wrinkle height is expected to be ~ 200 nm.
- In our previous study, we measured the wrinkle height of a sylgard substrate using AFM (atomic force microscopy) and the height was again in a range 200-300 nm (see Figure A; peak-to-peak height 500 nm). Although the material is different from the current paper (CY), it still gives us a rough estimate on the wrinkle height.

We added the following section.

page 7. In “Materials and Methods”

- added) new section “Wrinkle mechanics”

Comment 1-4:

What is the force threshold for wrinkle formation?

Figure A: The wrinkle height of a sylgard substrate using AFM (atomic force microscopy). An AFM image and phase contrast image taken at the same position are superimposed. The peak-to-peak height is ~ 500 nm as examined along the black line in the AFM image; and note that cells are plated on a square micropatterned region to minimize the movement of the cells during the image acquisitions, so that a corner of the square is imaged in the AFM/optical images.

Response to Comment 1-4:

As shown in Fig.3(b), there is no wrinkle generation when the average force is $\bar{f} \leq 10$ Pa.

Comment 1-5:

What is the sensibility of this approach? It is also unclear how this approach can resolve smaller forces that may arise at different angles than the prominent wrinkles.

Response to Comment 1-5:

The sensibility of the current approach is 10 Pa: the cells exhibit wrinkles when the average force \bar{f} is greater than 10 Pa as shown in Fig. 3(b).

Small forces can be recovered in our approach. As shown in Fig. 5, the force distributions are well predicted even when the place is far from the wrinkles. From the wrinkle geometry, our method predicts the landscape of force distributions and also how the forces decay with a distance.

Comment 1-6:

This approach uses PDMS for force measurement. What are the stiffness range that can be used to visualise wrinkles?

Response to Comment 1-6:

In the manuscript we have used the material with elasticity 5.4 kPa and as discussed in reply to the Comment 1-2 above, we have now extended the analyses to two additional substrates with stiffness 16.3 kPa (CY 1.0:1) and glass (see also Figure S3).

Moreover, as we have now summarized in the new content “Wrinkle mechanics”, not only the stiffness itself but the stiffness contrast between surface/bulk of the material is important. The wrinkle wavelength λ would be longer if the substrate E_p becomes stiffer compared to the bulk E_m . If we would like to observe 20 wrinkles for each cell with a size $\sim 100 \mu\text{m}$, the stiffness ratio should be $E_p/E_m \leq 1500$.

page 7. In “Materials and Methods”

- added) Note if one needs to observe 20 wrinkles for each cell with a size $\sim 100 \mu\text{m}$, the stiffness ratio should be $E_p/E_m \leq 1500$ from Eq. [9].

Comment 1-7:

Should a new Deep Learning model trained for each stiffness to be analysed?

Response to Comment 1-7:

It is recommended to do the training again for different stiffness in most of the cases, except limited cases described below.

Assume that the stiffness of oxidized layer is E_p^0 and bulk substrate is E_m^0 for our original experiment, while they are E_p^1 and E_m^1 for the experiment with new conditions. As we have described in the new section “Wrinkle mechanics”, conditions for the wrinkle generation would be identical if the stiffness ratio is the same for new experiment: $E_p^0/E_m^0 = E_p^1/E_m^1$. Therefore, only thing we need to modify is to multiply E^1/E^0 to the force that CNN predicted, for this case.

When the stiffness ratio is not fixed $E_p^0/E_m^0 \neq E_p^1/E_m^1$, CNN cannot directly predict the force distribution since the wavelength λ (Eq. [9] in the manuscript) and the critical strain for the wrinkle generation ε_c (Eq. [11]) is different for this condition. Although it might be still possible to estimate the strain considering the condition differences, it is still difficult to do the direct prediction as before.

We pointed out this statement in the new section.

page 7 “Wrinkle mechanics”

- If the substrate stiffness is different from the trained condition, the system needs to be trained once again with a new set of data. However, the trained system is still applicable when the stiffness ratio E_p/E_m of a new substrate is the same as the trained condition since the criteria for wrinkle generation would be identical, as shown in Eqs. [9] and [11]. By multiplying the ratio of Young’s modulus, we can estimate the force distribution just for this case.

Comment 1-8:

On the same point, Should a Deep learning model be trained for each cell type to be imaged? Can the authors demonstrate that this is not the case?

Response to Comment 1-8:

The wrinkle patterns are determined only from the traction distributions. Therefore, same principle can be applied to other cell lines unless the cells have comparable size with that of the

training data. When the cell size is not comparable, the model should be trained once again since wrinkle patterns with similar length scale might not be included in the training data.

Figure S6 shows the distribution of traction force for MEF (Mouse embryonic fibroblast), which has a comparable size with A7r5 cells (training data). Our machine learning approach well predicts the distribution from the wrinkle patterns.

page 6: in “Discussions”:

- Additionally, since the wrinkle patterns are determined only from the traction distributions, we expect the same principle can be applied to other cell lines unless the cells have comparable sizes with those used in the training data. When the cell size is not comparable, the model should be trained once again since wrinkle patterns with similar length scale might not be included in the training data. Figure S6 shows the distribution of traction force for MEF (Mouse embryonic fibroblast), which has a similar size with A7r5 cells (training data). Our machine learning approach well predicts the distribution from the wrinkle patterns.

In Supplemental materials:

added) Figure S6

Comment 1-9:

What is the resolution of the approach used here? The TFM maps used are of very low resolution. Can this approach also be used to resolve smaller details?

Response to Comment 1-9:

The spacial resolution of the force distribution is $3.44 \mu\text{m} \times 3.44 \mu\text{m}$.

Yes, we can change the spacial resolution to smaller value but there will be a trade-off with the accuracy. The resolution is limited by the PIV method, which is used to evaluate the substrate displacement. This method finds the displacement by image cross-correlations, and the prediction accuracy becomes lower by decreasing the reference image size. We use this spacial resolution to ensure the prediction accuracy.

Note that the spatial resolution is in a same order with other papers (for example, Tang *et al.*, PLOS Computational Biology, 2014), and this is a common resolution used in TFM.

Comment 1-10:

The authors show that they obtain excellent predictions using their approach. It is, however, unclear how these analyses were done as no details are provided. How many images were analysed? Were these images part of the training dataset used to train the Deep Learning network?

Response to Comment 1-10:

We wrote the detail in the section “Traction force prediction using GAN”, and we used $N = 252$ training images and 3 test images for the evaluation. The test images are not included during the training process. We repeated the test 5 times and the errors are evaluated with 15 total test images.

Comment 1-11:

While I do not think that the authors should discover new biological phenomena, they should demonstrate that this approach can be used to observe meaningful changes in force transmission. For instance, show that forces are reduced following a drug treatment.

Response to Comment 1-11:

- In our previous paper (Ref. 20: Nehwa *et al.*, BBRC, 2020), we used the current experimental setup (with the multi-well plate) to evaluate how the cellular force changes with 19 different drugs. By the drug treatment, the force, which is measured by the wrinkle length, became 20-130% compare to the control condition. This work clearly shows that the current approach can be utilized as a high-throughput system for force measurement. Note: In Nehwa *et al.*, we use the machine learning system just to extract the wrinkles (step2 in the present work), and the evaluation of force distribution (step 3) is newly developed for the present work.
- We add a new figure, Fig. 4, to show that the geometrical feature, whether the wrinkles are clustered or dispersed, can be used to estimate the relative strength of the cellular force and the force isotropy.

page 3, “Simultaneous measurement of wrinkles and traction forces”:

- added) Figure 4
- added) Table 1
- added) Topological features of the wrinkle also give us insights into the force distributions. As shown in Fig. 4(a), some cells exhibit wrinkles in a single region (top) while the others show several separated regions of wrinkles (bottom). By categorizing the cell images (total 34 images) into two types as clustered patterns (9 images) and dispersed patterns (25 images), we summarize the difference in the force distributions in Figs. 4(b) and (c): the average force \bar{f} and force isotropy I are significantly larger (\bar{f} : $p < 0.01$, I : $p < 0.05$) for clustered pattern, where isotropy is evaluated as $I = |f_p/f_p^{\min}|$ where f_p^{\min} is the smaller eigenvalue of S_{ij} . The wrinkles are generated by the pairwise forces that are transmitted via focal adhesions. When the wrinkles are dispersed, cells tend to be elongated, as seen in the pictures, and we can hypothesize that cells might not be strong enough to generate continuous wrinkles. The current result suggests that the wrinkle topological features have rich information on the force distributions.
- added) Finally, the correlations between cellular properties and characters of contractile forces are summarized in table 1. Since each variable has moderate positive correlations (~ 0.5) with all other variables, it can be concluded that the cells are circular when the size is larger, and both force magnitude and isotropy increase with the cell size.

Comment 1-12:

The discussion is very much a summary of the study itself. I would recommend that the authors also discuss the limitations of the approach described here and its possible uses.

Response to Comment 1-12:

Thank you for this suggestion. We have added discussion of the current limitations and possible applications of our approach to the Discussion section in the revised manuscript.

page 6, “Discussion”:

- added) The relationship between morphology and biological functions of living creatures has long been an intense subject of research. This topic has been investigated at the individual cellular level as well, in which cellular contractile forces were implicated in diverse functions including proliferation, differentiation, apoptosis, and tumorigenesis. Given its complicated nature, however, the whole relationship associated with cellular forces remains fully understood. In this regard, the technology described here has a potential to significantly advance research in this field as it allows for easier acquisition of the equivalent of TFM data. Indeed, we found the tendency of circular cells being stronger and higher in the magnitude and isotropy of the contractile force, respectively, compared to elongated ones. Thus, WFM will also be available, in addition to drug screening as we discussed below, for extensively probing how the force of cells is related to their functions including maintenance of morphological phenotypes.

It is instructive to discuss limitations and advantages of the proposed approach. In this paper we have presented data for a stiffness range 5.4 – 16.3 kPa. In principal, it is recommended to do the training again for different stiffness in most of the cases, except limited cases described below. As we have described in the Methods section “Wrinkle mechanics”, conditions for the wrinkle generation would be identical if the stiffness ratio is the same for new experiment: $E_p^0/E_m^0 = E_p^1/E_m^1$. Therefore, only thing we need to modify is to multiply E^1/E^0 to the force that CNN predicted, for this case.

When the stiffness ratio is not fixed $E_p^0/E_m^0 \neq E_p^1/E_m^1$, CNN cannot directly predict the force distribution since the wavelength λ (Eq. [9] in the manuscript) and the critical strain for the wrinkle generation ε_c (Eq. [11]) is different for this condition. Although it might be still possible to estimate the strain considering the condition differences, it is still difficult to do the direct prediction as before. Additionally, since the wrinkle patterns are determined only from the traction distributions, we expect the same principle can be applied to other cell lines unless the cells are not much bigger than those used in the training data. When the cell size is not comparable, the model should be trained once again since wrinkle patterns with similar length scale might not be included in the training data. Figure S6 shows the distribution of traction force for MEF (Mouse embryonic fibroblast), which has a similar size with A7r5 cells (training data). Our machine learning approach well predicts the distribution from the wrinkle patterns.

Comment 1-13:

Fiji software is not referenced.

Response to Comment 1-13:

We have cited Fiji software in our latest version.

Comment 1-14:

It would be useful to provide the actual images used to train and generated by the GAN. Not just images with force vectors.

Response to Comment 1-14:

We have uploaded the training data to an open-access workplace, at https://github.com/Minatsukiyoshino/Wrinkle_force_microscopy in the data folder.

REVIEWER #2:

The authors in this manuscript combine the well-recognized method of TFM with the much older described wrinkling imaging on silicon surfaces to propose an easier method to measure traction exerted by the cells on a deformable substratum using machine learning. The goal is to propose a much simpler method based on only phase contrast microscopy imaging. For this, authors seeded cells on silicon coated with fluorescent beads to allow performing on the same field traction force and wrinkling imaging. Then based on TFM data they trained neuronal network to extract traction fields.

Comment 2-1:

Although I am not in a position, as biologist, to evaluate what is new here compared to ref 18 and 19.

Response to Comment 2-1:

The biggest advance in this paper is the quantitative measurement of force distribution from the surface wrinkle. Previous works such as [Ref 18 - Burton and Taylor, 1997; Ref 26 - Fukuda *et al.*, 2017] provided an estimated of the force **magnitude** using an analytical relation that predicts a dependence of the contractile traction force magnitude on the wrinkle length. Although the approach was successful in giving rough estimations of the force change in a glance, it has a disadvantage in predicting the force **distribution**, which is important to understand the cellular mechanotransduction and morphologies. Due to the complex force-geometry relation of the wrinkle, Ref 18 can only predict the approximate force directions. Our present method overcame this disadvantage by letting the system learn the force-geometry relation.

Moreover, in terms of the physics of wrinkling there has been a long-standing debate on the force-wrinkle length dependence: While some earlier works (Ref 18) predicted a linear relationship, a rigorous analytical treatment of Ref 36 [Cerdea and Mahadevan, PRL, 2002] showed that the relationship should be quadratic. As shown in Figure 3(c) of our manuscript, the experimental measurements indeed show a rather linear relation between the traction force and the wrinkle length, which validates a more recent theoretical prediction based on far-from threshold theory of wrinkling [Davidovitch *et al.*, PNAS, 2011] and is consistent with the experiments on droplets on Polystyrene films [Huang *et al.*, Science 2007].

Comparison with Ref. 19 [Li et al., BBRC, 2019]: Ref 19 is our previous work, and the wrinkle extraction using the machine learning is proposed (Note that the wrinkle extraction method is utilized in “Step2: Wrinkle extraction” in the current work). The current work is a drastic improvement from the previous work, and we can now predict the stress distribution from the microscope images.

We stress this point in the introduction as follows.

page 1, in “Introduction”:

- added) Although the geometrical information of wrinkles (Groenewold, 2001; Beningo and Wang 2002; Cerda and Mahadevan, 2003), such as wavelengths, would give an estimation in the force magnitude and direction, the geometry is still not enough to predict the local force distribution in a sub-cellular scale that is important to understand the cellular mechanotransduction and morphologies.

page 3, in “Simultaneous measurement of wrinkles and traction forces”:

- added) While earlier work predicted a linear relationship between the traction force and the wrinkle length, analytical treatment showed that the relationship should be quadratic. The linear relation that is measured here validates a more recent theoretical prediction based on far-from threshold theory of wrinkling and is consistent with the experiments on droplets on polystyrene films.

Comment 2-2:

I have some concerns about, the spatial resolution

Response to Comment 2-2:

The spacial resolution of the force distribution is $3.44 \mu\text{m} \times 3.44 \mu\text{m}$. The spacial resolution can be set to smaller value but there will be a trade-off with the accuracy. The resolution is limited by the PIV method, which is used to evaluate the substrate displacement. This method finds the displacement by image cross-correlations, and the prediction accuracy becomes lower by decreasing the reference image size. We use this spacial resolution to ensure the prediction accuracy. Note that this spatial resolution is in a same order with other papers (for example, Tang *et al.*, PLOS Computational Biology, 2014), and this is a common resolution used in TFM.

Comment 2-3:

the sensitivity of the methods

Response to Comment 2-3:

As shown in Fig. 3(c), the cells exhibit wrinkles when the average force \bar{f} is greater than 10 Pa. Therefore, the forces can be predicted via the wrinkles for $\bar{f} \geq 10 \text{ Pa}$.

Comment 2-4:

it possible or not application to cell layers

Response to Comment 2-4:

Yes, we can apply this framework to predict the force of cell layers, but we need to do the training again with a dataset of cell layers.

The cell mechanics are different for single cells and cell layers: for example, there are no cell-cell junctions and cell-cell force transmissions for single cells. We tried to predict the force of cell layers with a system that was trained with the single cell dataset, but the prediction was not good compared to the one we reported in the paper. We are planning to report a system to predict the force of cell layers in our next work.

Comment 2-5:

the fact that silicon wrinkling modifying the layer in 3D may affect TFM extraction

Response to Comment 2-5:

The wrinkle generation would only have small error ($\sim 5\%$) on the strain estimation. Assume that there is a wrinkle with a shape $\zeta(x) = A \cos(2\pi x/\lambda)$ where A is the amplitude and λ is the

wavelength. By assuming that the amplitude is smaller than the wavelength ($A/\lambda \ll 1$) [Ref 22: Groenewold, Phys. A: Stat. Mech its Appl., 2001], the arc length of a single wave ℓ can be given as

$$\ell = \int_0^\lambda \sqrt{1 + \left(\frac{d\zeta(x)}{dx}\right)^2} dx \sim \int_0^\lambda \left(1 + \frac{2\pi^2 A^2}{\lambda^2} \sin^2(2\pi x/\lambda)\right) dx = \lambda \left(1 + \frac{\pi^2 A^2}{\lambda^2}\right).$$

This equation indicates that there would be a excess length change $\pi^2 A^2/\lambda$ when a segment with length ℓ shrinks to λ . As we discussed in a new section “Wrinkle mechanics”, the parameters are $\lambda \sim 3 \mu\text{m}$ and $A \sim 0.2 \mu\text{m}$ in our experiments. Using the parameters, the excess length $\pi^2 A^2/\lambda$ change compared to the wavelength λ is given as $\pi^2 A^2/\lambda^2 \approx 0.04$. Therefore, there is only partial effect on the strain estimation with the wrinkle generation. We added new section as follows.

page 7. In “Materials and Methods”

- added) new section “Wrinkle mechanics”

Comment 2-6:

Also, as authors say, this would be mostly interesting for screening, high throughput, but no proof of principle for its application in this field is provided.

Response to Comment 2-6:

In our previous paper (Ref. 20: Nehwa *et al.*, BBRC, 2020), we used the current experimental setup (with the multi-well plate) to evaluate how the cellular force changes with 19 different drugs. By the drug treatment, the force, which is measured by the wrinkle length, became 20-130% compare to the control condition. This work clearly shows that the current approach can be utilized as a high-throughput system for force measurement. Note: In Nehwa *et al.*, we use the machine learning system just to extract the wrinkles (step2 in the present work), and the evaluation of force distribution (step 3) is newly developed for the present work.

We added the following sentence.

In Introduction:

For instance, the force measurements with dozens of different drugs can be done simultaneously by implementing the wrinkle assay to a multi-well plate (Nehwa *et al.*, BBRC, 2020).

REVIEWER #3:

Li et al. provide an elegant method to extract quantitative absolute traction force measurements from wrinkle images using deep learning approaches. The absolute measure of forces is typically done using Traction Force Microscopy (TFM), which requires the imaging of fluorescent bead displacement which can then be converted into absolute forces from the knowledge of the mechanical properties of the substrate. An associated method uses the observation of substrate buckling (leading to the observation of wrinkles) which only requires the acquisition of brightfield or phase contrast of the cells' substrate but is less quantitative with respect to absolute force measurements. Here, the authors generate a paired dataset of phase contrast images of wrinkles and TFM data, the latter being able to generate force maps. Then they used this data to train a conditional GAN neural network to predict force maps from such phase contrast images. This has the advantage to only require the acquisition of phase contrast images without the need for a reference image (as it typically required by TFM). The authors show nice quantifications of the performance of the approach on test dataset that show a broad agreement with ground truth and demonstrate the validity of the method.

Comment 3-1:

Deep learning constitutes a great approach to perform complex transformation of data such as wrinkle images into force maps. The authors rightly explain that the information is there but may be difficult to extract in a quantitative and spatially resolved manner as TFM does. What I am not sure about is how it compares to previous efforts to convert wrinkle images into force measures. The authors mention that wrinkle length and direction constitute two measures that relate to amplitude and direction of the forces but has there been any past efforts to convert these into force maps using deep learning or otherwise? The authors should discuss this in a bit more details, perhaps in introduction, and if no work has ever intended to do this quantitatively successfully, that will only strengthen the case for using deep learning and the present method.

Response to Comment 3-1:

There is an attempt to only roughly evaluate the force **magnitude** using a simplified relation that the wrinkle length increases linearly with the contractile force (Ref. 18: Burton and Taylor, Nature, 1997) as we wrote in the introduction, but there is no attempt for quantitative conversion to the "actual" force **distribution**, to the best of authors' knowledge. Due to the complex force-geometry relation of the wrinkle, note that Ref. 18 could only predict the approximate force directions. Our present method overcame this disadvantage by letting the system learn the force-geometry relation, and thus we believe our method provides a huge advance over the previous studies, in obtaining the actual cellular force magnitude and distribution.

We added the following sentence to stress this point.

page 1, "Introduction":

- added) Although the geometrical information of wrinkles (Groenewold, 2001; Beningo and Wang 2002; Cerda and Mahadevan, 2003), such as wavelengths, would give an estimation in the force magnitude and direction, the geometry is still not enough to predict the local force distribution in a sub-cellular scale that is important to understand the cellular mechanotransduction and morphologies.

Comment 3-2:

The authors quantify errors compared to ground truth TFM data from the ensemble distribution of force magnitudes and angles. They show an agreement of the best performing method (GAN from wrinkles) within about 30% and mention that more training data would improve that. The angular errors are within 20 degrees. I would like to see a little more characterisation of these errors as they form the basis of the method and would help understand the caveats compared to TFM (here considered as gold standard). So here, I think it would be useful to see the actual distributions of force amplitudes and force angles from both predictions and ground truth to show precision and potential biases. This could also be briefly discussed in my opinion.

Response to Comment 3-2:

Thank you very much for your interesting suggestion, in response to which we add new figure S4. As shown in the figure, there is a good agreement between the ground truth and the prediction. Although the angle has small deviation, the prediction still well capture the distribution of the ground truth.

In Supplemental materials:

added) Figure S4

Comment 3-3:

Additionally, since the approach is meant to spatially resolve the forces, it would also be useful to show spatial error maps (difference or root square error, RSE, or similar) of both magnitude and angle for the test datasets, as is commonly done for validation of deep learning producing images. This would also potentially highlight issues of where the errors come from mostly and how they relate to certain spatial features. This would be a nice complement to the correlation curve shown in Fig. 5a.

Response to Comment 3-3:

We add new figures S2 in the appendix. As expected, the error is large at the image edge while it is relatively small at center.

In Supplemental materials:

added) Figure S5

Comment 3-4:

I also have a concern about generalisation of the work. Although the authors show nicely that there is a decent agreement with TFM from the test data. It looks to me that all the dataset (both training and test) were acquired on the same day from the same dish. I would like to see whether the approach would generalise to a couple of test datasets acquired on a different day from a different dish, that were not present in the training data. This would constitute the ideal test dataset here. Biological variability as well as variability of how the substrate may be made can cause variability that may throw off the model and produce poor quality predictions. This is not uncommon.

Response to Comment 3-4:

63 data is collected from the experiments of five different days and seven different dishes that are independently prepared. Since the training/test data were randomly picked from the data, our approach is generalized enough for different days/dishes (Note: 15 test data consist of

experiments of four different days with five different dishes). We mentioned this point as follows.

p. 7: Materials and Methods “Step 3: Prediction of traction force based on GAN-based system”

added) Note that 63 images are acquired from experiments on five different days and seven different dishes.

Comment 3-5:

Reproducibility is also an issue here since neither the data nor the code for Deep Learning has been made available freely and directly. This is to me an important aspect that’s missing and essential for transparency.

Response to Comment 3-5:

We have uploaded the code and the data to an open-access workplace, at https://github.com/Minatsukiyoshino/Wrinkle_force_microscopy.

Comment 3-6:

- Box plots in Fig. 5 are not defined.
- Temporal information of the time-course data shown in the movies are not indicated anywhere.
- Typo: “plasma-irradiated silocone”

Response to Comment 3-6:

- Description for the plot is added in Fig. 6 (previously Fig. 5).
- We wrote the time-interval between the frames in the movie caption.
- We have changed “plasma-irradiated silocone” to “plasma-irradiated silicone”

REVIEWERS' COMMENTS:

Reviewer #1 (Remarks to the Author):

The authors have addressed all my concerns. I recommend publication.

Reviewer #2 (Remarks to the Author):

I see that I shared my concerns with the two other reviewers. I see also that the authors address most of them in their rebuttal. They also corrected their manuscript taking into account some of them. However, I think that there is still place for improvement in adding these precisions and corrections in the manuscript itself.

This concerns: modifications in the text concerning responses to 1.4 (need to state in the text the sensitivity of the methods), 1.5, 1.9 (response needs to be stated in the manuscript), 1.10 (idem), 1.11 (the statement of similarities and novelties of the work compared to ref 20 need to be reported in the manuscript), same thing for 2.1, 1;12, the first added paragraph is way to general and not needed, discussion should be centered to novelty (comment 1.11) and limitation (precision, reproducibility, etc.).

I do not understand the first paragraph of the rebuttal and the rational to remove some data set?

Reviewer #3 (Remarks to the Author):

The authors have satisfactorily addressed my concerns with the right level of additional details and I am happy with publication of this manuscript.

The only point requiring modification is in Fig. S5, addressing my comment 3-3: can the author please show a panel with a few (3 or 4 maybe) representative examples of Error maps of SINGLE images with their corresponding Bright-field and ground truth images next to them, rather than the average of 15 test images.

I am concerned that the averaging would eliminate the feature-related errors and more interesting aspects of the performance of the methods.

Reply to reviewer's comments

We gratefully acknowledge the constructive comments and suggestions of the referees and the editor. The responses to each reviewer's comments are listed below.

REVIEWER #1:

The authors have addressed all my concerns. I recommend publication.

Response to Reviewer #1:

Thank you very much.

REVIEWER #2:

I see that I shared my concerns with the two other reviewers. I see also that the authors address most of them in their rebuttal. They also corrected their manuscript taking into account some of them. However, I think that there is still place for improvement in adding these precisions and corrections in the manuscript itself.

Comment 2-1:

modifications in the text concerning responses to 1.4 (need to state in the text the sensitivity of the methods), 1.5, 1.9 (response needs to be stated in the manuscript), 1.10 (idem), 1.11 (the statement of similarities and novelties of the work compared to ref 20 need to be reported in the manuscript) same thing for 2.1, 1.12, the first added paragraph is way to general and not needed, discussion should be centered to novelty (comment 1.11) and limitation (precision, reproducibility, etc..).

Response to Comment 2-1:

Thank you for your suggestion, but some of the statements are already included in the manuscript. We would like to answer to your requests separately as follows:

- Comment 1.4 and 1.5: We stated in the manuscript 10 Pa is the sensitivity of our current method.
Page 2, line 108: "*The wrinkle extincts when the mean traction in a image is less than 10 Pa, which is comparable to the noise level or the resolution of the current TFM.*"
- Comment 1.9: We added a new sentence as follows:
Added) Page 4, line 392: The spatial resolution of the force distribution is $3.44 \mu m \times 3.44 \mu m$.

- Comment 1.10: We wrote the detail in the section “Traction force prediction using GAN” as follows:

Page 5, line 437: *“Note that we used $N = 252$ training image sets and 3 test images for the evaluation. The total error is calculated by averaging the error of 15 test images, which are obtained by repeating the evaluation 5 times with randomly selected different test images.”*

- Comment 1.11: We add new sentences to show the common and different aspects between our previous work (ref 20) and the current work.

Added) Page 5, line 420: *Note that the procedure until Step 2 was already developed and utilized in our previous papers (Li *et al.* 2020, Nehwa *et al.* 2020), and GAN-based force estimation is newly presented in this work.*

- Comment 2.1 and 1.12: We believe we already added the novelty/limitation of method in the discussion:

Page 3, line 273: *“It is instructive to discuss limitations and advantages of the proposed approach. In this paper we have presented data for a stiffness range 5.4 – 16.3 kPa. In principal, it is recommended to do the training again for different stiffness in most of the cases, except limited cases described below. As we have described in the Methods section “Wrinkle mechanics”, conditions for the wrinkle generation would be identical if the stiffness ratio is the same for new experiment: $E_p^0/E_m^0 = E_p^1/E_m^1$. Therefore, only thing we need to modify is to multiply E^1/E^0 to the force that CNN predicted, for this case. When the stiffness ratio is not fixed $E_p^0/E_m^0 \neq E_p^1/E_m^1$, CNN cannot directly predict the force distribution since the wavelength λ (Eq. [9] in the manuscript) and the critical strain for the wrinkle generation ε_c (Eq. [11]) is different for this condition. Although it might be still possible to estimate the strain considering the condition differences, it is still difficult to do the direct prediction as before. Additionally, since the wrinkle patterns are determined only from the traction distributions, we expect the same principle can be applied to other cell lines unless the cells are not much bigger than those used in the training data. When the cell size is not comparable, the model should be trained once again since wrinkle patterns with similar length scale might not be included in the training data. Figure S6 shows the distribution of traction force for MEF (Mouse embryonic fibroblast), which has a similar size with A7r5 cells (training data). Our machine learning approach well predicts the distribution from the wrinkle patterns.”*

We also added a section “Statistics and Reproducibility” to describe the precision and reproducibility.

Added) Page 5, line 533: Section “Statistics and Reproducibility”

Comment 2-3:

I do not understand the first paragraph of the rebuttal and the rational to remove some data set?

Response to Comment 2-3:

We removed the data to show the robustness/reliability of our method. In the first submitted version, we included same cells at different time (few hours difference; 20 cells) in order to increase the number of the training data. Including same cells have a potential to overestimate the performance of our GAN system, since there is a possibility that we pick the same cells with different time for training/test images; for example, in training data: cell A at time I, in test

data: cell A at time II. We decided to remove those cells to avoid this risk, and we believe that this modification increased the robustness and reliability of our method.

REVIEWER #3:

The authors have satisfactorily addressed my concerns with the right level of additional details and I am happy with publication of this manuscript.

Comment 3-1:

The only point requiring modification is in Fig. S5, addressing my comment 3-3: can the author please show a panel with a few (3 or 4 maybe) representative examples of Error maps of SINGLE images with their corresponding Bright-field and ground truth images next to them, rather than the average of 15 test images. I am concerned that the averaging would eliminate the feature-related errors and more interesting aspects of the performance of the methods.

Response to Comment 3-1:

Following your suggestion, we added new figure S6 to show spatial errors for 3 different cells. Large errors tend to locate at the image edge, or places with small forces.